# Learning a Metric for Multimodal Medical Image Registration without Supervision Based on Cycle Constraints

**DOI:** 10.3390/s22031107

**Published:** 2022-02-01

**Authors:** Hanna Siebert, Lasse Hansen, Mattias P. Heinrich

**Affiliations:** Institute of Medical Informatics, Universität zu Lübeck, 23538 Lübeck, Germany; hansen@imi.uni-luebeck.de (L.H.); heinrich@imi.uni-luebeck.de (M.P.H.)

**Keywords:** image registration, cycle constraint, multimodal features, self-supervision, rigid alignment

## Abstract

Deep learning based medical image registration remains very difficult and often fails to improve over its classical counterparts where comprehensive supervision is not available, in particular for large transformations—including rigid alignment. The use of unsupervised, metric-based registration networks has become popular, but so far no universally applicable similarity metric is available for multimodal medical registration, requiring a trade-off between local contrast-invariant edge features or more global statistical metrics. In this work, we aim to improve over the use of handcrafted metric-based losses. We propose to use synthetic three-way (triangular) cycles that for each pair of images comprise two multimodal transformations to be estimated and one known synthetic monomodal transform. Additionally, we present a robust method for estimating large rigid transformations that is differentiable in end-to-end learning. By minimising the cycle discrepancy and adapting the synthetic transformation to be close to the real geometric difference of the image pairs during training, we successfully tackle intra-patient abdominal CT-MRI registration and reach performance on par with state-of-the-art metric-supervision and classic methods. Cyclic constraints enable the learning of cross-modality features that excel at accurate anatomical alignment of abdominal CT and MRI scans.

## 1. Introduction

Medical image registration based on deep learning methods has gathered great interest over the last few years. Yet, certain challenges, especially in multimodal registration, need to be addressed for learning based approaches, as evident from the recent MICCAI challenge *Learn2Reg* [1]. In order to avoid an elaborate comprehensive annotation of all relevant anatomies and to avoid label bias, unsupervised, metric-based registration networks are widely used for intramodal deep learning based registration [2,3].

However, this poses an additional challenge for multimodal registration problems, as currently no universal metric has been developed and a trade-off has to be made between using local contrast-invariant edge features such as NGF, LCC, and MIND or more global statistical metrics like mutual information. Metric-based methods also entail the difficulty of tuning hyperparameters that balance similarity weights (ensuring similarity between fixed image and warped moving image) and regularisation weights (ensuring plausible deformations).

Ground truth deformations for direct supervision are only available when using synthetic deformation fields. The now very popular FlowNet [4] estimates deformation fields between pairs of input images from a synthetically generated dataset that has been obtained by applying affine transformations to images. However, for medical applications, synthetic deformations have been deployed for monomodal image registration [5,6,7]. Alternatively, label supervision that primarily maximises the alignment of known structures with expert annotations could be employed [2,8,9]. This leads to improved registration of anatomies that are well represented, but can introduce a bias and deteriorating performance for unseen labels.

On the one hand, the focus of supervised approaches on a limited set of labelled structures may be particularly inadequate for diagnosis of a pathology that cannot be represented sufficiently in the training data. Using metric supervision, on the other hand, has little potential to improve upon classical algorithms that employ the same metric as similarity terms during optimisation. With efficient (parallelised) implementations, adequate runtimes of less than a minute have recently been achieved for classical algorithms.

Learning completely without metric or label supervision, self-supervision, would remedy the aforementioned problems and enable the development of completely new registration methods and multimodal feature descriptors without introducing annotation or engineering biases.

Self-supervision approaches have been used in medical and non-medical learning based image processing tasks. Recently, a self-supervised approach for learning pretext-invariant representations for object detection has outperformed supervised pre-training in [10]. By minimising a contrastive loss function, the authors construct image representations that are invariant to image patch pertubation, similar to the representation of transformed versions of the same image and differ from representations of other images. In [11], semantic features have been learned with self-supervision in order to recognise the rotation that has been applied to an image given four possible transformations as multiples of 90 degrees. The learned features have been useful for various visual perception tasks. For rigid registration between point clouds, an iterative self-supervised method has been proposed in [12]. Here, partial-to-partial registration problems have been addressed by learning geometric priors directly from data. The method comprises a keypoint detection module which identifies points that match in the input point clouds based on co-contextual information and aligns common keypoints. For monomodal medical image registration, in [13] spatial transformations between image pairs have been estimated in a self-supervised learning procedure. Therefore, an image-wise similarity metric between fixed and warped moving images is maximised in a multi-resolution framework while the deformation fields are regularised for smoothness.

In [14], cycle-consistency in time is used for learning visual correspondence from unlabelled video data for self-supervision. Their idea is to obtain supervision for correspondence by tracking backward and then forward, i.e., along a cycle in time, and use the inconsistency between the start and end points as the loss function. For image-to-image translation, a cycle-consistent adversarial network approach is introduced in [15]. The authors use a cycle consistency loss that induces the assumption that forward and backward translation should be bijective and inverse of each other. Another approach that addresses inconsistency is introduced in [16] for medical image registration. It uses information from a complete set of pairwise registrations, aggregates inconsistency, and minimizes the group-wise inconsistency of all pairwise image registrations by using a regularized least-squares algorithm. The idea to measure consistency via registration cycles for monomodal medical image data has been used in [17] that estimates forward and reverse transformation jointly in a non-deep-learning approach and [18] using registration circuits to correct registration errors. In [19], a monomodal unsupervised medical image registration method that trains deep neural network for deformable registration is presented using CNNs with cycle-consistency. This approach uses two registration networks that process the two input images as fixed and moving images inversely to each other and gives the deformed volumes to the networks again to re-deform the images to impose cycle-consistency.

Previous deep learning based registration work has often omitted the step of rigid or affine registration, despite its immense challenges due to often large initial misalignments. Image registration challenges such as [1] provide data that has been pre-aligned with help of non-deep-learning-based methods, whereas the challenge’s image registration tasks are then often addressed with deep learning based methods. Rigid transformation is often the inital step before performing deformable image registration, and only few works [20] investigate deep learning techniques for this step. As evident from the CuRIOUS challenge [21], so far no CNN approach was able to learn a rigid or affine mapping between multimodal scan pairs (MRI and ultrasound of neurosurgery) with an adequate robustness. Besides that, no label bias can occur with rigid alignment. Hence, a learning model for large linear transformations is of great importance.

### Contributions

In order to avoid the difficulty of choosing a metric for multimodal image registration, we propose a completely new concept. For learning multimodal features for image registration, our learning method requires neither label supervision nor handcrafted metrics. It extends upon research that successfully learned monomodal alignment through synthetic deformations, but transforms this concept to multimodal tasks without resorting to complex modality synthesis.

The basic idea of our novel learning based approach is illustrated in Figure 1. It relies on geometric instead of metric supervision. In this work

We introduce a cycle based approach including cycles that for each pair of CT and MRI scans comprise two multimodal transformations to be estimated and one known synthetic monomodal transformation.We restrict ourselves to rigid registration and aim to learn multimodal registration between CT and MRI without metric supervision by minimising the cycle discrepancy.We use a CNN for feature extraction with initially separate encoder blocks for each modality followed by shared weights within the last layers.We use a correlation layer without trainable weights and a differentiable least squares fitting procedure to find an optimal 3D rigid transformation.We created to the best of our knowledge the first annotated MRI/CT dataset with paired patient data that are made publicly available with manual segmentations for liver, spleen, left and right kidney.

Our extensive experimental validation on 3D rigid registration demonstrates the high accuracy that can be achieved and the simplicity of training such networks.

## 2. Materials and Methods

We introduce a learning concept for multimodal image registration that learns without metric supervision. Therefore, we propose a method to learn with the help of a self-supervised learning procedure using three-way cycles. For our registration models, the architectural design consists of modules for feature extraction, correlation, and registration. Implementation details, open source code and trained models can be found at github.com/multimodallearning/learningwithoutmetric (accessed on 8 March 2021).

### 2.1. Self-Supervised Learning Strategy

Our deep learning based method learns multimodal registration without using metric supervision. Instead, it is based on geometric self-supervision by minimising the cycle discrepancy created through a cycle consisting of two multimodal transformation and one monomodal transformation. The basic cycle idea is illustrated in Figure 1: Initially, a fixed image (Image 1) and a moving image (Image 2) exist. The transformation R_21_ is unknown and is to be learned by our method. In each training iteration, we randomly deform the moving image (Image 2) by applying a known random transformation R_23_ and hereby obtain a synthetic image (Image 3). By bringing the individual transformations into a cycle, the minimisation problem of
(1)$$|R_{23}\cdot R_{31}-R_{21}|\to \min$$
can be derived. We chose to minimise the discrepancy as given in Equation (Equation 1) instead of minimising the difference between the transformation combination $R_{23}\cdot R_{31}\cdot R_{12}$ and the identity transformation Id with $|R_{23}\cdot R_{31}\cdot R_{12}-\mathrm{Id}|\;\to \;\min$ in order to avoid that our method only learns identity warping. For optimisation, we use the mean squared error loss function to minimise the cycle discrepancy between the two flow fields generated by the transformation matrices R_21_ and $R_{23,31}=R_{23}\cdot R_{31}$.

As we restrict our model to rigid registration, we create the synthetic transformations R_23_ by randomly initialising rigid transformation matrices with values that are assumed to be realistic from an anatomical point of view.

The advantages of our learning concept are manifold. First, in comparison to supervising the learning with a known similarity metric and regularisation term, the need for balancing a weighting term is removed and the method is applicable to new datasets without domain knowledge. Second, it enables multimodal learning, which is not feasible using synthetic deformations in conjunction with image-based loss terms (cf. [6]). Third, it avoids the use of domain discriminators as used, e.g., in the CycleGAN approach [15,22], which usually requires a large set of training scans with comparable contrast in each modality and may be sensitive to hyper-parameter choices.

On first sight, it might seem daring to use such a weak guidance. While it is clear that once suitable features are learned the loss term enables convergence, since the cycle constraint is fulfilled. Yet to initiate training towards improved features, we primarily rely on the power of randomness (by drawing multiple large synthetic deformations) and explorative learning. In addition, the architecture contains a number of stabilising elements: a patch-based correlation layer computation, outlier rejection and least squares fitting, that are described in detail below in Section 2.3 and Section 2.4.

### 2.2. Training Pipeline

We apply our self-supervised learning strategy in the training procedure by going through the same steps in each training iteration as visualised in Figure 2: First, a random transformation matrix R_23_ is generated and applied on the moving image in order to obtain the synthetic image. Then, moving and fixed image are passed through feature extraction, correlation layer and transformation computation module to obtain the transformation matrix R_21_. The same step is also performed for fixed and synthetic image to obtain R_31_. After this, R_23_ and R_31_ are combined to obtain R_23,31_. Finally, the mean squared error of the deformations calculated with help of R_21_ and R_23,31_ is determined. The individual modules for this training pipeline are described in more detail in the following Section 2.3 and Section 2.4.

### 2.3. Architecture

The architecture used for our registration method comprises three main components for feature extraction, correlation, and transformation computation.

We chose to use a CNN for feature extraction with initially separate encoder blocks for each modality and shared weights within the last few layers. These features are subsequently fed into the correlation layer, which has no trainable weights and whose output could be directly converted into displacement probabilities. Our method employs a robust and differentiable least squares fitting to find an optimal 3D rigid transformation subject to outlier rejection. Figure 3 visualises the procedure for feature extraction, correlation, and computation of the rigid transformation matrix that is used for registration.

For our feature extraction CNN, we use a Y-shaped architecture (cf. Figure 3) [9] starting with a separate network part for each of the two modalities (ModalityNet), which takes the respective input and passes it through two sequences with a structure of 2×.

(Strided) 3D convolution with a kernel size of three and padding of one;3D instance normalisation;leaky ReLU.

The two convolutions of the first sequence are non-strided and output eight feature channels. The first convolution of the second sequence has a stride of two and doubles the number of feature channels to 16, whereas the second convolution of the second sequence is non-strided and keeps the number of 16 feature channels. Whereas the size of the input dimensions are preserved within the first convolution sequence, the strided convolution within the second sequence leads to a halving of each feature map dimension. The output of the ModalityNets are passed into a final module with shared weights (SharedNet), which finalises the feature extraction by applying two sequences of the same structure as used for the separate ModalityNets. Here, the first sequence comprises non-strided convolutions that output 16 feature channels while keeping the spatial dimensions as output by the ModalityNets. The first convolution of the second sequence has a stride of two leading to another halving of the spatial dimension’s sizes and doubles the number of feature channels to 32. The second convolution of the second sequence is non-strided and keeps the number of 32 feature channels. The output of the SharedNet is given to a 1×1×1-convolution providing the final number of 16 feature channels followed by a Sigmoid activation function. As we use correlation and transformation estimation techniques without trainable weights, our model only comprises 80k trainable parameters within the feature extraction part.

### 2.4. Correlation and Transformation Computation

As suggested in previous research [3,4], the use of a dense correlation layer that explores a large number of discretised displacements at once is employed to capture larger deformations robustly. This way the learned features are used to define a sum of squared differences cost function akin to metric learning [23].

Similar to [24], which operates directly on input image pairs and uses normalised cross correlations (NCC), we use a block-matching technique to find correspondences between the fixed features and a set of transformed moving features. We correlate the obtained features by calculating patch-wise the sum of squared differences (SSD) and extract points with high similarity of a coarse grid with a spacing of 12 voxel. The extracted grid points are used to define point-wise correspondences to calculate the rigid transformation matrix with a robust (trimmed) least squares fitting procedure.

For the correlation layer, we choose a set of 11×11×11 discrete displacements with a capture range of approx. 40 voxel in the original volumes. After calculating the sum-of-squared-differences cost volume, we sort the obtained SSD costs and reject the 50% of the displacement choices that entail the highest similarity costs. We apply the Softmax function on the remaining displacement choices to obtain differentiable soft correspondences. While we use this differentiable approach to estimate regularised transformations within a framework that comprises trainable CNN parameters, the learned features could also be used for other optimisation frameworks [9].

The displacement candidates output by the Softmax function are added to the coarse moving grid points. In a least squares fitting procedurce comprising five iterations, the final rigid transformation matrix that serves for transformation of the moving image is determined. The best-fitting rigid transformation can be found by computing the singular value decomposition $S=U\Sigma V^{T}$ with the matrix $S=X^{T}Y^{T}$ (*X*: centered fixed grid points $x_{i}$, *Y*: centered moving grid points with added displacement candidates $y_{i}$) and the orthogonal matrices *U* and *V* obtained by the singular value decomposition. This leads to the rotation
(2)$$Q=V\left(\begin{array}{ccccc} 1 & \; & & \; & \\ \; & 1 & \; & & \\ \; & \; & ... & \; & \\ \; & \; & & 1 & \\ \; & \; & & \; & \det(VU^{T}) \end{array}\right)U^{T}$$
and the translation
(3)$$t=\bar{y}-Q\bar{x}$$
with $\bar{x}$ being the mean values for fixed grid points and $\bar{y}$ the mean moving grid points with added displacement candidates.

This way, the rigid transformation matrices R_21_ and R_31_ are determined. To combine the synthetic transformation R_23_ and the predicted transformation R_31_ matrix multiplication is used yielding R_23,31_. The transformation matrices R_21_ and R_23,31_ are used to compute the affine grids that are then given to the MSE loss function during training and the affine grid computed by R_21_ is used for warping during inference to align the moving image to the fixed image.

This approach has the advantage of being very compact with only 80k parameters ensuring memory efficiency and fast convergence of training. The multimodal features learned by our model are generally usable for image alignment and can be given to various optimisation methods for image registration once trained with our method.

## 3. Experiments and Results

Our experiments are performed on 16 paired abdominal CT and MR scans from collections of The Cancer Imaging Archive (TCIA) project [25,26,27,28]. We have manually created labels for four abdominal organs (liver, spleen, left kidney, right kidney), which we use for the evaluation of our methods. Apart from a withheld test set, they are publicly released for other researchers to train and compare their multimodal registration models. The pre-processing comprises reorientation, resampling to an isotropic resolution of 2 mm and cropping/padding to volume dimensions of 192×160×192.

To increase the number of training and testing pairs and model realistic variations in initial misalignment we augment the scans with 8 random rigid transformations each that on average reflect the same Dice overlap (of approx. 43%) as the raw data. All models are trained for 100 epochs with a mini-batch size of 4 in less than 45 min each using ≈8 GByte GPU memory.

The weights of the CNN used for feature extraction (FeatCNN) are trained for 100 epochs using the Adam optimiser with an initial learning rate of 0.001 and an cosine annealing scheduling.

### 3.1. Comparison of Training Strategies

We compare three different strategies to train our FeatCNN in a two-fold cross-validation:FeatCNN + Cycle Discrepancy (ours): Our proposed self-supervised cycle learning strategy;FeatCNN + MI Loss: Learning with metric-supervision using Mutual Information (MI) as implemented by [29];FeatCNN + NCC^2^ Loss: Learning with metric-supervision using squared local normalised cross correlations (NCC^2^) [24,30];FeatCNN + Label Loss: Supervised learning with label supervision.

All methods share the same settings for the correlation layer and a trimmed least square transform fitting (with five iterations and 50% outlier rejection). Hyperparameters were determined on a single validation scan (#15) for cyclic training and left unaltered for all other experiments. The same trainable FeatCNN comprising the layers as described in Section 2.3 is used to train with our Cycle Discrepancy Loss, MI, NCC^2^, and Label Loss. For correlation, we chose to extract corresponding grid points within a grid with a spacing of 12 voxels and use patches with a radius of 2 to patch-wise calculate the SSD. We use a displacement radius of 4 and discretise the set of displacements possibilities for the correlation layer with a displacement step (resp. voxel spacing) of 5. To adjust the smoothness of the soft correspondences, the costs obtained by SSD computation are multiplied by a factor of 150 when given to the Softmax function. As the soft-correspondences are needed for differentiability only during training, we increase this factor to 750 for inference.

For our cycle discrepancy method, we create the synthetic transformation matrices R_23_ by randomly initialising them with values that are assumed to be realistic from an anatomical point of view. Therefore, the maximum rotation is ±23° and the maximum translation ±42 voxel (which equals 84 mm for our experiments) in every image dimension.

The results demonstrate a clear advantage of our proposed self-supervised learning procedure with an average Dice of 72.3% compared to the state-of-the-art MI metric loss with 68.14% and NCC^2^ Loss with 68.1%, which is suitable for multimodal registration due to its computation involving small local neighbourhoods [24] (see Table 1 for qualitative and Figure 4 for quantitative results). This result comes close to the theoretical upper bound of our model trained with full label supervision with 79.55%.

### 3.2. Comparison of Inference Strategies and Increased Trainset

To further enhance our method, we extend it by a two-level warping approach during inference. Therefore, we present our model the input moving and fixed image to warp the moving image and then apply our model to the resulting warped moving image and the fixed image again. For both warping steps, we set a displacement radius of 7 voxel and a grid spacing of 8 voxel. For the first warping step, we use a displacement discretisation of 4 voxel and refine this hyperparameter to 2 voxel for the second warping step.

Moreover, as our dataset is quite small and our method does not require labels, when considering an application scenario where a number of MR/CT scan pairs have to be aligned offline, a fine-tuning of the networks on this test data would be feasible. Therefore, we aim to further increase the performance of our method with training on all available paired CT and MR scans without splitting the dataset.

In Table 2 we compare the results of single-level and two-level warping as well as the cross-validation results and the results when training on the whole available image data. We compare the results achieved by our method with the results achieved using the rigid image registration tool *reg_aladin* of NiftyReg [24] applied to the image pairs used without the symmetric version and one registration level.

Introducing a second warping step increased our cross-validation results by more than 4% points. When training without a withheld testset, we achieved further improvements by another 3% points. These results are on a par with the results of state-of-the-art classic method NiftyReg- *reg_aladin*.

## 4. Discussion

In this work, we presented a completely new concept for multimodal feature learning with application to 3D image registration without supervision of labels or handcrafted metrics. We introduced a new supervision strategy that is based on synthetic random transformations (two across modality and one within) that form a triangular cycle. Minimising the two multimodal transformations in such a cycle constraint avoids singular solutions (predicting identity transforms) and enables the learning of large rigid deformations. Through explorative learning, we are able to successfully train modality independent feature extractors that enable highly accurate and fast multimodal medical image alignment by minimising a cycle discrepancy in training. We also created the first public multimodal 3D MRI/CT abdominal dataset with manual segmentations for validation. To the best of our knowledge our work is also the first deep learning model for robustly estimating large misalignments of multimodal scans.

Despite the very promising results, there are a number of potential extensions that could further improve our concepts. The idea of incremental learning and predicting more useful synthetic transformations to improve detail alignment could be considered and has already shown potential in preliminary 2D experiments.

While the gap between training and test accuracy is relatively small due to the robust architectural design, further fine-tuning would be applicable at test time (since no supervision is required) with moderate computational effort. Combining hand-crafted domain knowledge with self-supervised learning might further boost accuracy. Similarly, domain adaptation through adversarial training could be incorporated to explicitly model the differences of modalities. While the gap between training and test accuracy is relatively small due to the robust architectural design, further fine-tuning would be applicable at test time (since no supervision is required) with moderate computational effort.

## 5. Conclusions

With our method, we were able to improve over the use of handcrafted metric-based losses by using synthetic three-way cycles. By minimising the cycle discrepancy, we are able to learn multimodal registration between CT and MRI without metric supervision. We created a robust method to estimate large rigid transformations that is differentiable in end-to-end learning. Our method is able to successfully perform intra-patient abdominal CT-MRI registration that outperforms state-of-the-art metric-supervision.

## Figures and Tables

**Figure 1 sensors-22-01107-f001:**
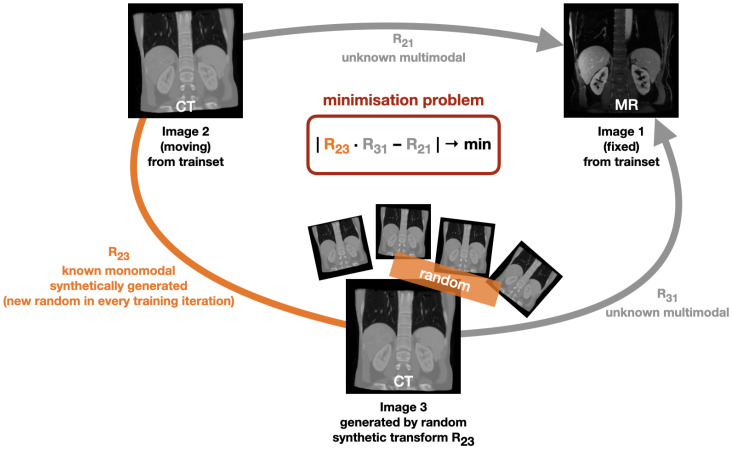
Our proposed self-supervised learning concept for multimodal image registration aiming to minimise a cycle discrepancy. In every training iteration, another (known) random transformation matrix R_23_ is used to generate a synthetic image. Like this, a cycle consisting of two unknown multimodal transformations (with the transformation matrices R_21_ and R_31_) and a known monomodal transformation (with the transformation matrix R_31_) is obtained, leading to the minimisation problem of $|R_{23}\cdot R_{31}-R_{21}|\to \min$ that is used for learning.

**Figure 2 sensors-22-01107-f002:**
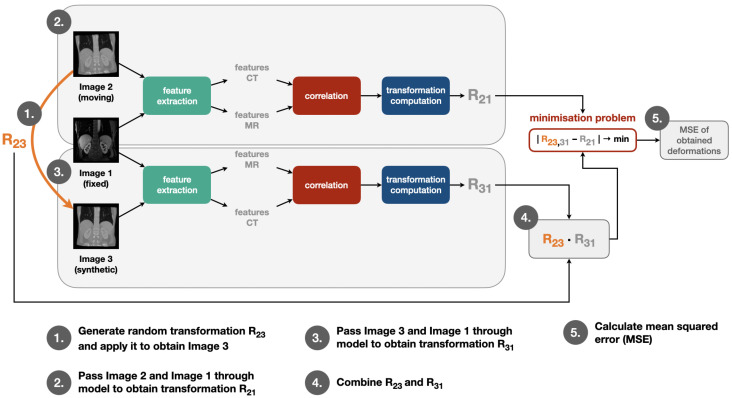
Pipeline to train our registration model: A random transformation matrix R_23_ is generated and used to obtain the synthetic image. The pair of moving and fixed image as well as the pair of synthetic and fixed image are passed through feature extraction, correlation layer and transformation computation module (see following Section 2.3 and Section 2.4) to obtain the transformation matrices R_21_ and R_31_. Then, R_23_ and R_31_ are combined to obtain R_23,31_. As a final step, the mean squared error (MSE) of the deformations calculated with help of R_21_ and R_23,31_ is determined.

**Figure 3 sensors-22-01107-f003:**
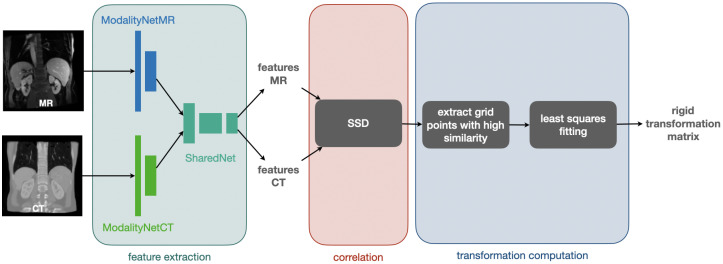
The process of feature extraction, correlation, and computation of the rigid transformation matrix: A CNN is used for feature extraction starting with a separate network part for each modality (ModalitiyNetMR and ModalityNetCT) followed by a module with shared weights (SharedNet). The obtained features are correlated by calculating patch-wise the sum of squared differences (SSD). Subsequently, grid points with high similarity are extracted and used to define point-wise correspondences to calculate the rigid transformation matrix with a least squares fitting.

**Figure 4 sensors-22-01107-f004:**
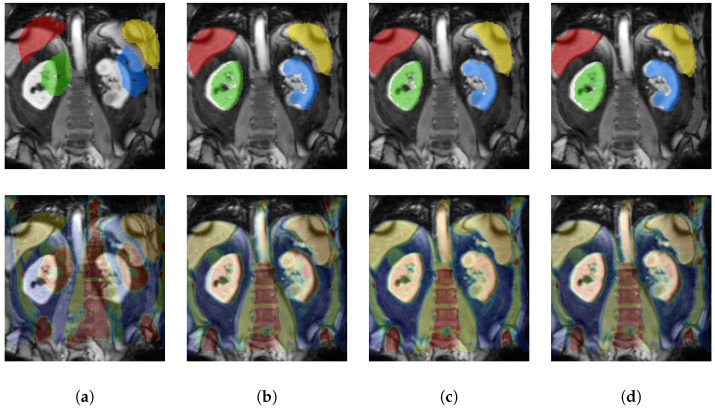
Qualitative results of our proposed cycle discrepancy approach FeatCNN + Cycle Discrepancy (**c**). We visualise the comparison to initial (**a**) before warping as well as to the methods FeatCNN + MI Loss (**b**) and FeatCNN + Label Loss (**d**) (coronal slices). The **top row** shows the fixed MRI and (warped) moving labels. The **bottom row** visualizes the (warped) moving CT and a jet colourmap overlay of the fixed MRI scan.

**Table 1 sensors-22-01107-t001:** Results for our cross-validation experiments: Dice scores listed by anatomical structures of our 3D experiments using FeatCNN for feature extraction and MI Loss, NCC^2^ Loss, Label Loss or our Cycle Discrepancy for training.

	Liver	Spleen	Lkidney	Rkidney	Mean
initial	59.32	36.90	36.59	37.02	42.46
	± 14.03	± 19.49	± 19.53	± 22.08	± 18.78
FeatCNN + MI Loss	75.07	63.17	69.86	64.46	68.14
	± 9.38	± 22.13	± 26.34	± 29.45	± 21.92
FeatCNN + NCC^2^ Loss	75.08	61.09	72.19	64.04	68.10
	± 12.22	± 23.69	± 27.51	± 31.76	± 23.80
FeatCNN + Cycle Discrepancy	77.95	69.89	70.18	71.85	72.30
	± 8.16	± 16.00	± 24.34	± 34.40	± 20.75
FeatCNN + Label Loss	81.24	73.84	83.15	79.97	79.55
	± 8.75	± 18.32	± 26.62	± 33.59	± 21.82

**Table 2 sensors-22-01107-t002:** Results for our experiments comparing single-level and two-level warping approach as well as cross-validation and training without withheld data: Dice scores listed by anatomical structures of our experiments using Cycle Discrepancy for training.

	Liver	Spleen	Lkidney	Rkidney	Mean
initial	59.32	36.90	36.59	37.02	42.46
	± 14.03	± 19.49	± 19.53	± 22.08	± 18.78
cross-validation	77.95	69.89	70.18	71.85	72.30
1 warp	± 8.16	± 16.00	± 24.34	± 34.40	± 20.75
cross-validation	80.71	72.12	79.33	74.65	76.68
2 warps	± 9.33	± 17.08	± 26.06	± 36.91	± 22.34
trained without withheld data	81.04	71.11	76.27	76.49	76.23
1 warp	± 8.22	± 18.03	± 24.25	± 32.64	±20.88
trained without withheld data	81.85	76.77	79.81	80.17	79.65
2 warps	± 0.58	± 13.64	± 24.52	± 34.65	± 20.25
NiftyReg	83.97	76.55	79.83	79.26	79.90
*reg_aladin*	± 6.19	± 12.00	± 7.12	± 37.55	± 15.15

## Data Availability

Our experiments are performed on abdominal CT and MR scans from collections of The Cancer Imaging Archive (TCIA) project [25,26,27,28]. Material is available under TCIA Data Usage Policy and Creative Commons Attribution 3.0 Unported License. Material has been modified for direct usage in registration and deep learning algorithms: We have reorientated the data, resampled it to an isotropic resolution of 2 mm, and used cropping and padding to achieve voxel dimensions of 192 × 160 × 192. We have also manually created segmentations for liver, spleen, left kidney, and right kidney. The results shown here are in whole or part based upon data generated by the TCGA Research Network: http://cancergenome.nih.gov/ ( accessed on 7 January 2021).

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
