# Peer review of "Learning a Metric for Multimodal Medical Image Registration without Supervision Based on Cycle Constraints"

_sensors, 2022, doi:10.3390/s22031107_

Round 1

Reviewer 1 Report

In this paper, the authors introduce a deep learning-based method for multimodal 3D image registration without the supervision of labels or handcrafted metrics. The proposed method uses synthetic three-way (triangular) cycles that for each pair of images comprise two multimodal transformations to be estimated and one known synthetic monomodal transform. The network is optimized by minimizing the cycle discrepancy and adapting the synthetic transformation to be close to the real geometric difference of the image pairs. Experiments on 16 paired abdominal CT and MR scans demonstrate the effectiveness of the proposed method.

This paper is well-motivated and the proposed method is technically sound. The idea of cycle consistency has been extensively studied for 2D image registration in both supervised and unsupervised settings. The author introduces this idea to multimodal 3D image registration and formulates an interesting unsupervised 3D registration method. 

The authors should give implementation details in section 3. For example, the authors failed to explain the rotation and translation range of random transformation R_23.    In Fig. 3, "SSD" is used without definition. It’s better to give the definition in the caption.  

Reviewer 2 Report

This paper proposes a medical image registration method based on cycle constraints. Experiments demonstrate the effectiveness of the proposed method when compared with several models. Overall, the paper is well presented. I have some comments as follows:

  1. In Figures 1 and 2, it would be better to replace the patterns of triangle and circle with medical images used in the experiments.
  2. As the authors mentioned that the global statistical metrics and local contrast-invariant edge features. However, only mutual information is used in the experiments. How about other local contrast-invariant edge features?
  3. It is recommended to review the self-supervision in other computer vision domains, such as Self-supervised learning of pretext-invariant representations (CVPR-2020), Unsupervised single image deraining with self-supervised constraints (ICIP-2019), etc.
  4. Some symbols lack explanations, such as the Id, Equations (2) and (3). Please check it.

Round 2

Reviewer 2 Report

The authors have addressed my comments.